# Structural basis for activation, assembly and membrane binding of ESCRT-III Snf7 filaments

Shaogeng Tang[1,2], W Mike Henne[1,2†], Peter P Borbat[3,4], Nicholas J Buchkovich[1,2‡], Jack H Freed[3,4], Yuxin Mao[1,2], J Christopher Fromme[1,2], Scott D Emr[1,2*]

[1]Weill Institute of Cell and Molecular Biology, Cornell University, Ithaca, United States; [2]Department of Molecular Biology and Genetics, Cornell University, Ithaca, United States; [3]National Biomedical Center for Advanced Electron Spin Resonance Technology, Cornell University, Ithaca, United States; [4]Department of Chemistry and Chemical Biology, Cornell University, Ithaca, United States

**Abstract** The endosomal sorting complexes required for transport (ESCRTs) constitute hetero-oligomeric machines that catalyze multiple topologically similar membrane-remodeling processes. Although ESCRT-III subunits polymerize into spirals, how individual ESCRT-III subunits are activated and assembled together into a membrane-deforming filament remains unknown. Here, we determine X-ray crystal structures of the most abundant ESCRT-III subunit Snf7 in its active conformation. Using pulsed dipolar electron spin resonance spectroscopy (PDS), we show that Snf7 activation requires a prominent conformational rearrangement to expose protein-membrane and protein-protein interfaces. This promotes the assembly of Snf7 arrays with ~30 Å periodicity into a membrane-sculpting filament. Using a combination of biochemical and genetic approaches, both *in vitro* and *in vivo*, we demonstrate that mutations on these protein interfaces halt Snf7 assembly and block ESCRT function. The architecture of the activated and membrane-bound Snf7 polymer provides crucial insights into the spatially unique ESCRT-III-mediated membrane remodeling.

**\*For correspondence:** sde26@cornell.edu

**Present address:** [†] Department of Cell Biology, University of Texas Southwestern Medical Center, Dallas, United States; [‡] Department of Microbiology and Immunology, Pennsylvania State University College of Medicine, Hershey, United States

**Competing interests:** The authors declare that no competing interests exist.

## Introduction

The endosomal sorting complexes required for transport (ESCRTs) are membrane remodeling machinery that mediate diverse fundamental cellular processes, including the biogenesis of multivesicular body (MVB) during receptor down-regulation (*Katzmann et al., 2001*), enveloped virus budding (*Garrus et al., 2001*), cytokinesis (*Carlton and Martin-Serrano, 2007*), plasma membrane repair (*Jimenez et al., 2014*), nuclear pore complex assembly (*Webster et al., 2014*), and nuclear envelope reformation (*Olmos et al., 2015*; *Vietri et al., 2015*). Originally identified using yeast genetics, ESCRTs package ubiquitinated transmembrane proteins into intraluminal vesicles (ILVs) that bud into the interior of the late endosome, creating a MVB that ultimately delivers cargos into the yeast lysosome (vacuole). The ESCRT pathway achieves receptor sorting through an elaborate division of labor. Upstream ESCRT components, ESCRTs-0, I, and II, assemble into stable hetero-multimers to sort ubiquitinated cargo on the endosomal surface by binding ubiquitin and endosomal lipid, phosphatidylinositol 3-phosphate (PI(3)P). In addition, ESCRT-II sets the architecture and initiates the assembly of the ESCRT-III complex, which together with Vps4 is responsible for remodeling endosomal membranes (*Henne et al., 2011*; *Hurley and Hanson, 2010*).

ESCRT-III is a unique protein complex in that it is metastable and conformationally dynamic, forming hetero-oligomeric filaments of multiple subunits on membranes (*Saksena et al., 2009*; *Teis et al., 2008*). Its subunits are inactive monomers in the cytoplasm, which activate and assemble

**eLife digest** A cell constantly remodels its surface to adapt to its environment, as well as to replace old or damaged proteins. To achieve this, cell-surface receptors are taken inside the cell and delivered to organelles called endosomes, where a molecular machine called ESCRT governs the receptors' fate. Distinct ESCRT complexes remodel the endosomal membrane to form vesicle packages that encapsulate the receptor proteins. These vesicles bud off into the endosome, which is then targeted to another organelle called the lysosome where the receptor proteins are degraded.

If the vesicles are unable to make their deliveries, the resulting sustained receptor activity can lead to numerous developmental and neurodegenerative diseases, as well as cancer. Remarkably, the ESCRT machinery also plays critical roles during cell division and the release of the human immunodeficiency virus (HIV) from host cells.

Previous studies demonstrated that a particular ESCRT complex, called ESCRT-III, forms spiraling filaments on membranes to generate vesicles. However, how the individual components of ESCRT-III assemble into such filaments was a mystery. Now, Tang et al. have determined the first X-ray crystal structures of the main component of ESCRT-III, a polymer of the protein called Snf7, and thus uncovered how these membrane-bound Snf7 spirals assemble.

Using a combination of cell biology, genetics and biochemistry techniques, Tang et al. also demonstrated that the Snf7 structures are necessary for ESCRT-III to work correctly inside living cells. Despite this achievement, key questions remain. The main one is how the other subunits of ESCRT-III interact and work together to remodel the membrane to form the vesicle packages at the endosomes.

into spiraling polymers on endosomes to drive cargo sequestration, membrane invagination and constriction (*Buchkovich et al., 2013*; *Hanson et al., 2008*; *Henne et al., 2012*; *Wollert and Hurley, 2010*).

ESCRT-III is a hetero-polymer of four "core' subunits of Vps20, Snf7/Vps32, Vps24 and Vps2 (*Babst et al., 2002*), and 'accessory' subunits of Ist1, Did2/Vps46, Vps60 (*Rue et al., 2008*) and Chm7 (*Horii et al., 2006*). Although all ESCRT-III subunits share a common domain organization, each subunit appears to contribute a specific function. ESCRT-II directly engages Vps20 to trigger a sequential activation and ordered assembly of ESCRT-III subunits at endosomes (*Teis et al., 2010*). Vps20 nucleates the homo-oligomerization of the most abundant ESCRT-III subunit, Snf7, which then recruits Vps24 and Vps2 (*Teis et al., 2008*). Vps2 finally engages the Vps4 complex for ESCRT-III disassembly (*Lata et al., 2008b*; *Obita et al., 2007*), making individual subunits available for subsequent rounds of vesicle formation.

ESCRT-mediated membrane remodeling produces membrane curvature that pushes away from the cytoplasm, which is topologically opposite to that of the 'classical' clathrin and COP-I/II vesicle budding reactions. This unique membrane bending topology highlights an ancient and central role of the ESCRT machinery in cellular remodeling events. However, due to the relative instability and heterogeneity of ESCRT-III polymers, high-resolution structural studies have generally been problematic. Structural work on Snf7 in particular has been difficult, due to its ability to assemble readily into polymers that interfere with crystallization. Ultimately, atomic-resolution structural information is necessary to understand how ESCRT-III achieves ordered assembly and membrane remodeling in diverse cellular pathways.

Even with limited structural information, previous studies have revealed distinct regions of Snf7 critical to ESCRT function. Snf7 contains a highly structured 'core' domain of four α-helices (*Muziol et al., 2006*). The C-terminus, in contrast, is less structured, including an α-helix (α5) that folds back against the core domain *in cis* to mediate autoinhibition (*Lata et al., 2008a*), a microtubule interacting and transport (MIT)-interacting motif (MIM) for Vps4 recognition (*Obita et al., 2007*), and an α-helix (α6) for Bro1/Alix interaction (*McCullough et al., 2008*) (*Figure 1A*).

How is Snf7 activated to promote ESCRT-III assembly? Numerous studies indicate that ESCRT-III subunits are activated by intramolecular conformational changes that promote protein-protein interactions (*Henne et al., 2012*; *Lata et al., 2008a*; *Saksena et al., 2009*; *Schuh et al., 2015*;

*Shen et al., 2014*), but the structural basis for this is obscure. Available X-ray crystal structures of the autoinhibited Vps24 (*Muziol et al., 2006*) and Ist1 (*Bajorek et al., 2009*; *Xiao et al., 2009*) suggest that these conformational changes involve the disruption of intramolecular interactions between the basic N-terminus and the acidic C-terminus. Upon releasing this autoinhibition, Snf7 subunits assemble into higher-order protofilaments or spirals (*Cashikar et al., 2014*; *Hanson et al., 2008*; *Henne et al., 2012*; *Shen et al., 2014*) with a range of different morphologies and dimensions.

Here, we present two X-ray crystal structures that unravel the molecular mechanism governing Snf7 conformational activation and polymer assembly. By selectively removing its autoinhibitory C-terminus, we determine the first crystal structure of the Snf7 core domain in the active conformation at 1.6 Å resolution. Surprisingly, rather than adopting a rigid four-helix coiled-coil, the core domain undergoes a large-scale conformational rearrangement to extend into a highly elongated structure. This conformational change not only extends a cationic membrane-binding surface, but also exposes hydrophobic and electrostatic protein interacting surfaces for polymerization. *In vitro* reconstitution and pulsed dipolar electron spin resonance spectroscopy (PDS) demonstrate that full-length Snf7 adopts the same active conformation and assembles into ~30 Å periodic protofilaments on a near-native lipid environment. Using negative stain transmission electron microscopy (TEM) and quantitative flow cytometry, we further demonstrate that mutations on key protein interfaces halt Snf7

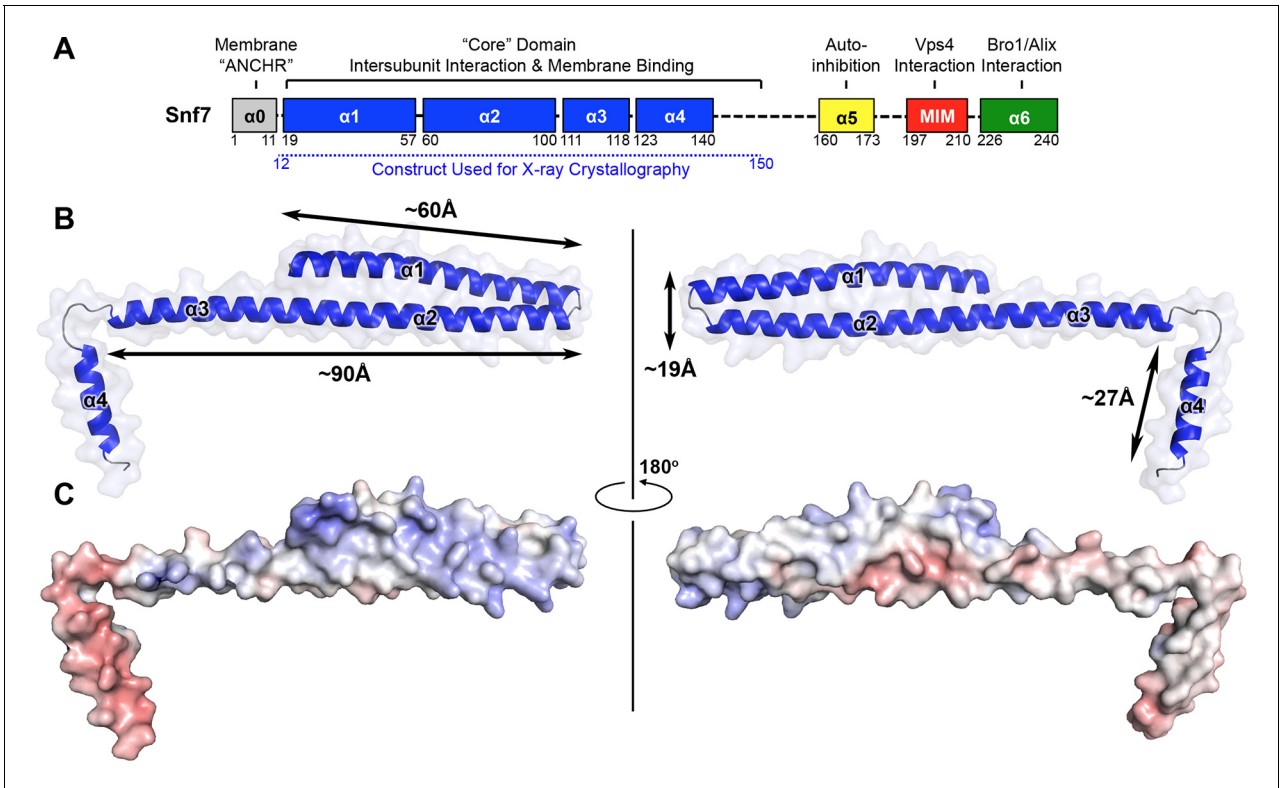

**Figure 1.** X-ray Crystal Structure of Snf7$^{core}$ (**A**) The domain organization of Snf7. The core domain used for X-ray crystallography is shown in blue. (**B**) Overlay of ribbon and space-filling models of the X-ray crystal structure of Snf7$^{core}$. (**C**) Electrostatic surface potential of Snf7$^{core}$ with positively charged regions in blue (+10kcal/e$^-$) and negatively charged regions in red (-10kcal/e$^-$). See also *Table 1*.

The following figure supplements are available for figure 1:

**Figure supplement 1.** Protein purification of Snf7$^{core}$

**Figure supplement 2.** 2Fc-Fo simulated-annealing composite-omit electron density maps contoured at 1.0$\sigma$ of Snf7$^{core}$ open conformations (**A**) A and (**B**) B.

**Figure supplement 3.** Superimposing Snf7$^{core}$ (blue) with (**A**) CHMP4B$^{\alpha1-\alpha2}$ (cyan) (PDB: 4ABM), with (**B**) CHMP3$^{\alpha1-\alpha4}$ (purple) (PDB: 3FRT), with (**C**) CHMP6$^{\alpha1}$ (red) (PDB: 3HTU) Snf7$^{core}$, and with (**D**) IST1$^{\alpha1-\alpha6}$ (grey) (PDB: 3FRR).

**Table 1.** Crystallographic Data Collection and Refinement Statistics

| | Snf7$^{core}$ | |
| --- | --- | --- |
| | Conformation A | Conformation B |
| Wavelength (Å) | 0.978 | 0.978 |
| Resolution range (Å) | 50 - 2.4 (2.49 - 2.40) | 50 - 1.6 (1.6 - 1.55) |
| Space group | $P2_1$ | $P2_1$ |
| Unit cell | $a = 29.5$Å $b = 52.2$Å $c = 54.5$Å $\alpha = 90°$ $\beta = 97.5°$ $\gamma = 90°$ | $a = 29.9$Å $b = 46.2$Å $c = 44.6$Å $\alpha = 90°$ $\beta = 98.5°$ $\gamma = 90°$ |
| Total reflections | 23263 (1946) | 73723 (6034) |
| Unique reflections | 6376 (612) | 16849 (1581) |
| Multiplicity | 3.6 (3.2) | 4.4 (3.8) |
| Completeness (%) | 97.99 (93.72) | 95.77 (90.65) |
| Mean I/sigma(I) | 8.04 (2.91) | 8.85 (1.35) |
| Wilson B-factor | 54.03 | 25.39 |
| $R_{merge}$ | 0.0884 (0.249) | 0.0782 (0.997) |
| $CC_{1/2}$ | 0.988 (0.968) | 0.995 (0.590) |
| CC* | 0.997 (0.992) | 0.999 (0.861) |
| $R_{work}$ | 0.259 (0.398) | 0.210 (0.330) |
| $R_{free}$ | 0.262 (0.533) | 0.225 (0.356) |
| Number of non-hydrogen atoms | 982 | 1097 |
| macromolecules | 975 | 992 |
| water | 7 | 105 |
| Protein residues | 123 | 125 |
| RMS(bonds) (Å) | 0.015 | 0.006 |
| RMS(angles) (°) | 1.24 | 0.81 |
| Ramachandran favored (%) | 95 | 99 |
| Ramachandran outliers (%) | 1.7 | 0 |
| Clashscore | 21.56 | 9.9 |
| Average B-factor | 91.1 | 39.7 |
| macromolecules | 91.2 | 38.9 |
| solvent | 69.2 | 47.2 |

assembly and block ESCRT function *in vivo*. Collectively, the molecular architecture of the activated and polymeric ESCRT-III Snf7 filament provides a detailed structural explanation for the mechanism underlying ESCRT-III-mediated membrane remodeling.

## Results

### X-ray crystal structure of Snf7$^{core}$

Despite reconstituting and visualizing ESCRT-III assembly with the resolution of TEM, it was unclear how Snf7 is conformationally activated, and how this activation coordinates the assembly of Snf7 polymers on membranes. To answer these questions, we sought to determine the structure of Snf7 at atomic resolution.

Because Snf7 intermolecular interactions rely primarily on core-core and core-membrane interactions (*Figure 1A*) (*Buchkovich et al., 2013*; *Henne et al., 2012*), we purified Snf7$^{core}$ to homogeneity (*Figure 1—figure supplement 1*). We then crystallized and solved X-ray crystal structures of

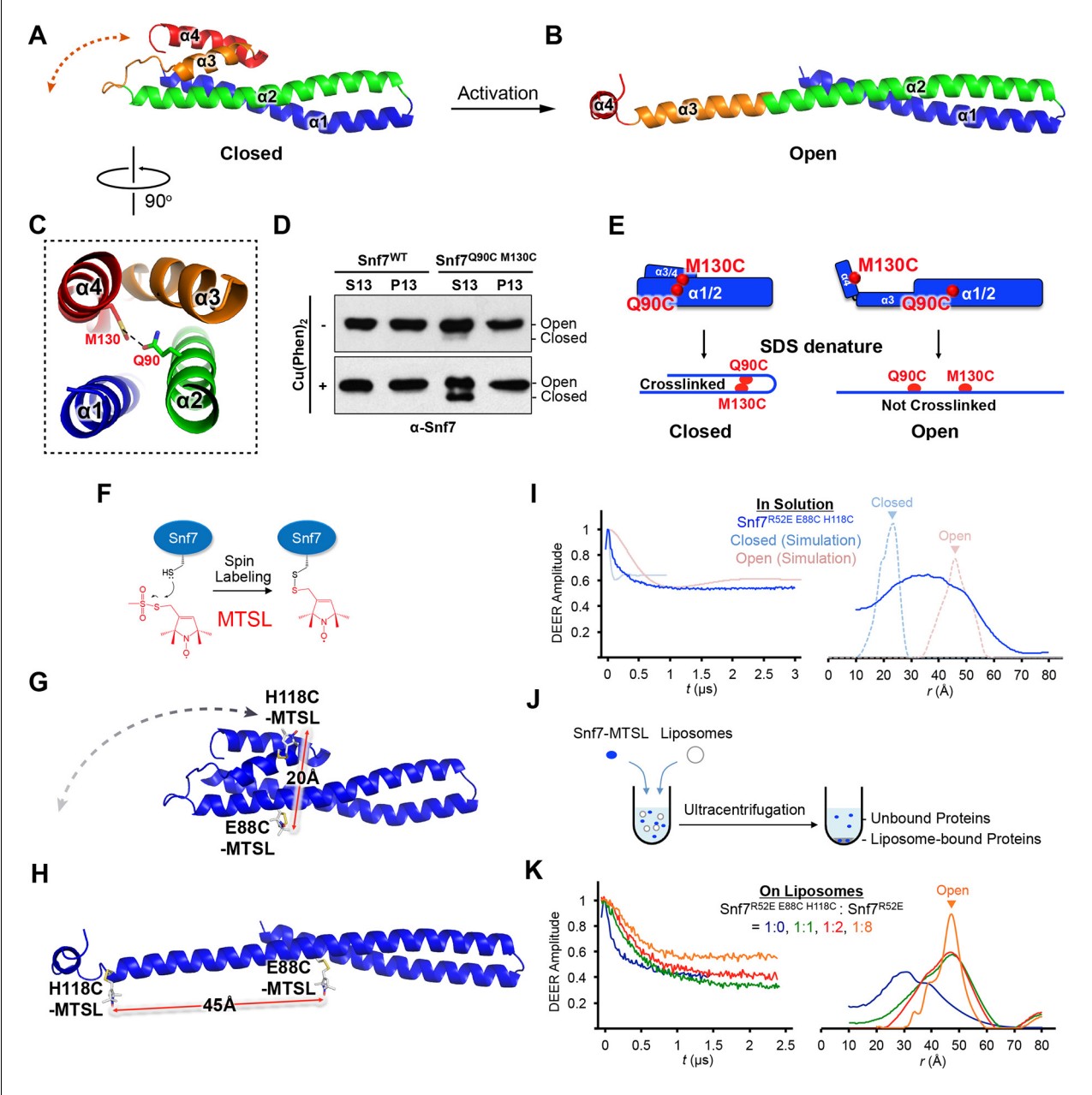

**Figure 2.** Conformational Rearrangement of Snf7 (A–B) Ribbon diagrams of (A) a homology model of closed Snf7[core] (*Henne et al., 2012*) and (B) the X-ray crystal structure of open Snf7[core]. (C) A close-up view of the side chain interaction between Gln90 and Met130. (D) Western blotting and subcellular fractionation of *snf7Δ* yeast exogenously expressing *SNF7* or *snf7[Q90C M130C]* with and without copper(II) 1,10-phenanthroline. (E) Schematic showing closed and open Snf7[core] with cysteines (red dots) before and after SDS-denaturing. (F) Snf7 site-directed spin-labeling with MTSL (red). (G–H) Distance between Glu88 and His118 of (G) closed and (H) open Snf7 shown in ribbon. (I and K) Time domain signals and distance distributions from DEER spectroscopy of (I) Snf7[R52E E88C H118C] in solution, and simulated closed and open Snf7[core E88C H118C] using *MMM*, and (K) Snf7[R52E E88C H118C]: Snf7[R52E] (1:0, 1:1, 1:2, and 1:8) with liposomes. (J) Schematic showing liposome sedimentation for DEER. MTSL-labeled Snf7 proteins (blue oval) and liposomes (grey circle).

The following figure supplements are available for figure 2:

**Figure supplement 1.** Conceptual model for the Mup1-pHluorin MVB sorting assay.

**Figure supplement 2.** Sequence alignments of Snf7 α2 and α4, with conserved Gln90 and Met130 shown in red, and quantitative MVB sorting data for *snf7Δ* yeast exogenously expressing *SNF7*, *snf7[Q90C]*, *snf7[M130C]*, and *snf7[Q90C M130C]*.

*Figure 2 continued on next page*

*Figure 2 continued*

**Figure supplement 3.** Time domain signals and distance distributions from DEER spectroscopy of full-length Snf7$^{R52E\ E88C\ H118C}$, Snf7$^{R52E\ H118C\ G140C}$ and Snf7$^{R52E\ E88C\ G140C}$.

Snf7$^{core}$ in two conformations at 1.6 Å and 2.4 Å resolutions, respectively. The structures were determined by molecular replacement using CHMP4B$^{\alpha1-\alpha2}$ (PDB: 4ABM) (*Table 1*, *Figure 1—figure supplement 2*). Although two conformations were determined, they share a similar overall tertiary structure with one notable exception discussed further below.

All previous ESCRT-III X-ray crystal structures adopt a canonical four α-helical core domain fold (*Bajorek et al., 2009*; *Muziol et al., 2006*; *Xiao et al., 2009*). When we superimposed our Snf7 structure with available ESCRT-III structures (*Figure 1—figure supplement 3*), we were surprised to note that Snf7$^{core}$ does not fold into four α-helices, but instead, it contains only three α-helices that pack into a highly elongated structure (*Figures 1B–C*). Although the α1/2 hairpin is relatively unchanged, α3 and α4 undergo large-scale structural rearrangements from the proposed autoinhibited ESCRT-III fold. α2 extends into a ~90 Å long α-helix combining the α2 and α3 segments that were distinct α-helices in previously defined ESCRT-III structures (*Figures 2A–B*). The angle of the flexible loop between α3 and α4 also changes, which enables α4 to position in different orientations relative to the α1-3 hairpin. Despite the conformational change, we designated this elongated α-helix as α2/3 to maintain a consistent numbering scheme for conserved ESCRT-III helices.

## Snf7 'opening' coupled with endosomal recruitment

Previous studies suggested that a 'closed' Snf7 becomes activated by the displacement of α5 away from the core domain (*Henne et al., 2012*; *Lata et al., 2008a*). Using a homology model of closed Snf7 (*Henne et al., 2012*) (*Figure 2A*), we identified close proximity between conserved residues Gln90 (α2) and Met130 (α4) in the four-helix coiled-coil (*Figure 2C*).

We applied a cysteine-based crosslinking strategy to directly monitor the conformational states of Snf7 *in vivo*. We mutated both Gln90 and Met130 to cysteines, and expressed this mutant in *snf7Δ* yeast (*Figure 2—figure supplements 1* and *2*). Since conformationally active Snf7 resides on endosomal membranes, we performed subcellular fractionation and collected the supernatant (S13) and the membrane-enriched pellet (P13) fractions. Western blotting analysis showed that Snf7$^{Q90C\ M130C}$ migrated to ~37 kDa, comparable to cysteineless Snf7. We then oxidized both fractions using copper(II) 1,10-phenanthroline. Strikingly, in the S13 fraction, ~50% of Snf7$^{Q90C\ M130C}$ migrated faster, indicating a conformationally closed Snf7 species. Notably, in the P13 fraction, the migration shift did not occur (*Figures 2D–E*). This is indicative of distinct conformations between the cytoplasmic and the endosome-bound states, and suggests that Snf7 on endosomal membranes adopts an open conformation in which α4 is displaced away from α2.

## Polymeric Snf7 adopts the open conformation

To investigate Snf7 activation at a structural level, we applied the PDS technique of double electron-electron resonance (DEER) and monitored full-length Snf7 in solution and bound to liposomes. As an approach to characterize protein conformations (*Borbat and Freed, 2007*; *Borbat and Freed, 2014*; *Jeschke, 2012*), PDS can provide distance constraints with a range of ~10–90 Å by measuring the magnitude of the dipolar coupling between spins of unpaired electrons in nitroxide spin labels (*Hubbell et al., 2000*). Snf7 assembles into spiraling protofilaments on membranes, presenting two challenges: (1) to characterize the conformational state of Snf7 building blocks; and (2) to determine the protofilament assembly from these structural elements.

To determine whether Snf7 activation induces the 'open' conformation we observed by X-ray crystallography, we selected two solvent-accessible residues, Glu88 (α2) and His118 (α3), predicted to be separated by a short distance of 20 Å in the closed state (*Figure 2G*), and an expected longer distance of 45 Å in the open state (*Figure 2H*). We labeled these two sites with a nitroxide spin label, MTSL (*Figure 2F*), and then obtained the distance distribution for the full length Snf7 in solution. The result showed a wide distance spread of ~15–50 Å (*Figure 2I*), corresponding to large amplitude motions of the spin labeled positions, but not a distinct closed or open state. Thus,

soluble Snf7 is structurally heterogeneous, suggesting that it is conformationally dynamic (*Figure 2— figure supplement 3*).

To map the active conformation of Snf7, we reconstituted spin-labeled full-length Snf7$^{R52E}$ polymers on lipid membranes, where R52E is a previously characterized activation mutant that induces Snf7 polymerization (*Henne et al., 2012*). We mixed the double-labeled Snf7$^{R52E\ E88C\ H118C}$ proteins with liposomes and collected the membrane-bound Snf7 polymers by ultracentrifugation (*Figure 2J*). Intriguingly, membrane-bound Snf7$^{R52E\ E88C\ H118C}$ produced a strong ~30 Å peak. We

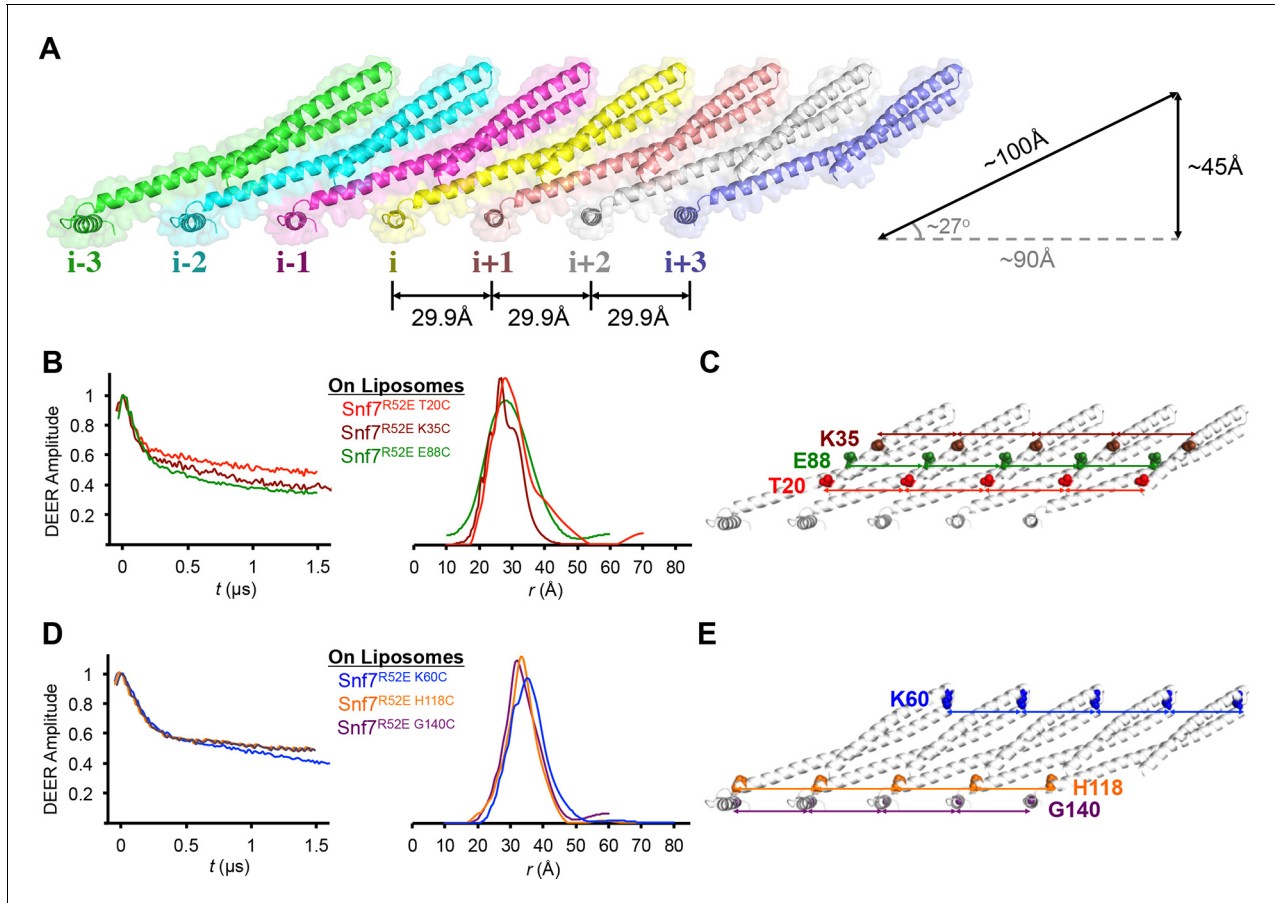

**Figure 3.** Membrane-bound Snf7 Protofilament with ~30 Å Periodicity (A) Overlay of ribbon and space-filling models of a 7-mer Snf7 protofilament with measured dimensions. (B and D) Time domain signals and distance distributions from DEER spectroscopy of (B) full-length Snf7$^{R52E\ T20C}$, Snf7$^{R52E\ K35C}$, and Snf7$^{R52E\ E88C}$ with liposomes, (D) full-length Snf7$^{R52E\ K60C}$, Snf7$^{R52E\ H118C}$, and Snf7$^{R52E\ G140C}$ with liposomes. (C and E) Schematic showing the spin label positions in a Snf7 protofilament.

The following figure supplements are available for figure 3:

**Figure supplement 1.** Time domain signals and distance distributions from DEER spectroscopy of full-length Snf7$^{R52E\ K60C\ A66C}$ in solution and full-length Snf7$^{R52E\ K60C\ A66C}$: Snf7$^{R52E}$ (1:0, 1:2) with liposomes, and schematic showing the locations of the spin label positions in a Snf7 protofilament.

**Figure supplement 2.** Time domain signals and distance distributions from DEER spectroscopy of full-length Snf7$^{R52E\ E88C\ H118C}$: Snf7$^{R52E}$ (1:0, 1:2.5, 1:4, 1:8) with liposomes and simulated Snf7$^{core\ E88C\ H118C}$: Snf7$^{core}$ (1:0, 1: ∞) polymers using *MMM*, and schematic showing the locations of the spin label positions in a Snf7 protofilament.

**Figure supplement 3.** Quantitative MVB sorting data for *snf7Δ* yeast exogenously expressing *SNF7*, *snf7$^{T20C}$*, *snf7$^{K35C}$*, *snf7$^{K60C}$*, *snf7$^{E88C}$*, *snf7$^{H118C}$*, *snf7$^{G140C}$*, *snf7$^{K60C\ A66C}$* and *snf7$^{E88C\ H118C}$*.

**Figure supplement 4.** Representative TEM images of recombinant full-length Snf7$^{R52E\ K35C}$, Snf7$^{R52E\ E88C}$, Snf7$^{R52E\ K60C\ A66C}$, and Snf7$^{R52E\ E88C\ H118C}$ labeled with MTSL.

also observed a significant population of distances at 40–50 Å, but diminished signal at ~20 Å (*Figure 2K*). We postulated that both the inter- and intra-subunit interspin distances contribute these signals. To isolate the intra-subunit interspin distance, we next produced magnetically diluted samples (*Borbat and Freed, 2007*; *Dzikovski et al., 2011*) by mixing double-labeled Snf7$^{R52E\ E88C\ H118C}$ with unlabeled Snf7$^{R52E}$ in ratios ranging from 1:1 to 1:8. We observed that the signal changed significantly up to 1:2 dilution, then less for the maximal 1:8 dilution (*Figure 2K* and *Figure 3—figure supplement 2*), showing approach to the infinite dilution limit. The data for the 1:8 dilution is characteristic of a single long distance of 45 Å with a moderate distance distribution, as expected for spin labels on an α-helix separated by 29 residues.

In summary, the reconstructed distance distributions are consistent with structural rearrangements that transform α2 and α3 into one continuous α-helix in the membrane-bound active conformation. As we did not observe short distances corresponding to the closed conformation, we conclude that only the open conformation is present in Snf7 polymers assembled on membranes. Therefore, the large-scale conformational rearrangement observed in the crystal structures is fully consistent with the PDS data of the full-length Snf7 conformations on the membranes.

## Membrane-bound Snf7 protofilaments exhibit a ~30 Å periodicity

While examining the arrangement of Snf7 molecules in the crystal lattice, we noted that multiple Snf7 protomers are arrayed into polymeric lattices, reminiscent of the protofilaments previously observed by TEM (*Henne et al., 2012*). Each of the ~100 Å long α1–3 hairpin tilts by ~27° and polymerizes into a ~45 Å diameter single protofilament, with each protomer exhibiting a repeat distance of ~30 Å (*Figure 3A*).

The spacing of protomers in the crystal is also in agreement with our DEER results of the full-length Snf7 protofilaments assembled on liposomes. We performed a series of DEER measurements on the single-labeled protein at several key positions. Specifically, we selected the middle of the α1-3 hairpin (Thr20, Lys35 and Glu88), and at both ends of α2/3 (Lys60 and His118), and the end of α4 (Gly140) to spin label Snf7. This allowed us to probe the interface between adjacent Snf7 protomers to establish their mutual orientation. Importantly, these cysteine-substituted Snf7 mutants are capable of assembling into protofilaments *in vitro* and do not impair MVB sorting *in vivo* (*Figure 3—figure supplement 3* and *4*).

Consistent with the extensive amount of inter-molecular contacts revealed in the Snf7 crystal, we observed moderately broad distance distributions, specifically at 28–32 Å for T20C, K35C and E88C (*Figures 3B–C*), and at 32–36 Å for K60C, H118C and G140C (*Figures 3D–E*). The modulation depths of the time-domain echo signals indicate a ~3-spin system, in agreement with the crystalline arrangement of Snf7, where each protomer has two neighboring protomers. The magnetic dilution (*Figure 3—figure supplements 1* and *2*) readily removed the intersubunit couplings, indicating that protofilaments do not make extensive contacts homogenous with each other.

Based on this series of single-cysteine DEER scanning and the double-cysteine magnetic dilution experiments, we conclude that Snf7 packing adopts a *parallel arrangement* in a single-layer array with a period of ~30 Å, and the reconstituted full-length Snf7 spirals on liposomes adopt a packing pattern similar to the Snf7$^{core}$ crystals. Thus, our X-ray crystal structures provide a foundation for in-depth study of the membrane-bound Snf7 polymer.

## Snf7 protomer interactions in the protofilament require two interfaces

In the Snf7 protofilament, the protomer (i) interacts with the next protomer (i+1) through both hydrophobic and electrostatic interactions (*Figures 4A–B*), burying ~1060 Å$^2$ of solvent-accessible surface area per protomer. The assembly of the extended α2/3 helix exposes a hydrophobic surface on α3, which was buried in the closed state. This enables the α2/3 helix of protomer (i) to interact with α2/3 of its neighboring protomer (i+1) (*Figure 4C*). Notably, Gln90, which interacts with Met130 *in cis* in the closed state, interacts with Met107 *in trans* in the open state.

To validate the interactions present in this hydrophobic interface, we performed site-directed mutagenesis and tested each mutant *in vivo* by an established quantitative MVB sorting assay (*Buchkovich et al., 2013*; *Henne et al., 2012*). This assay monitors the efficiency of fluorescence quenching after internalization and MVB sorting of Mup1-pH (the plasma membrane methionine transporter, Mup1, fused to the pH-sensitive GFP-derivative, pHluorin) (*Figures 4E* and *Figure 2—*

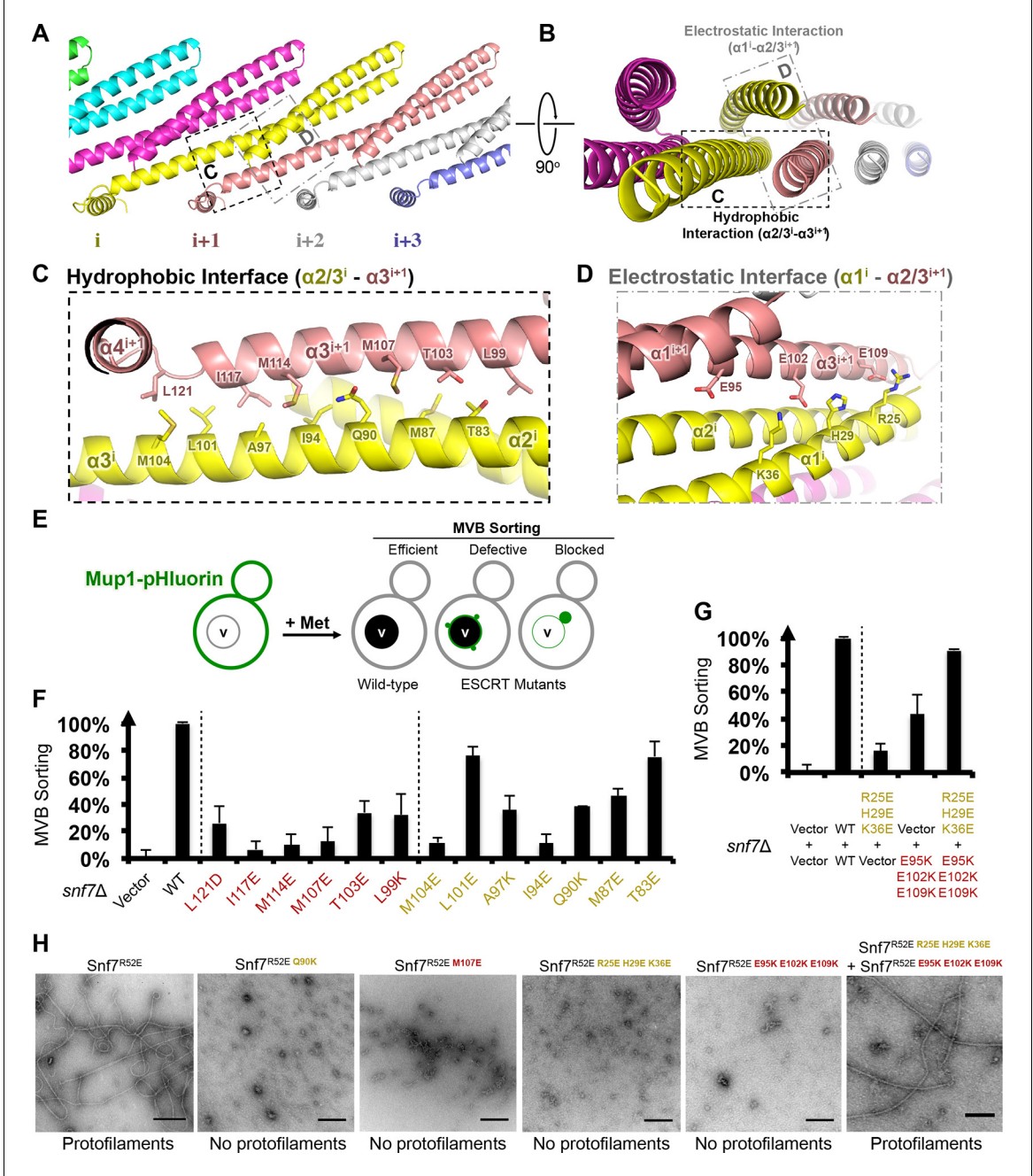

**Figure 4.** Hydrophobic and Electrostatic Interactions in a Snf7 Filament (**A–B**) Ribbon models of a Snf7 protofilament. The hydrophobic protein interface is shown in black dash-line and the electrostatic interface in grey dash-dot line. (**C–D**) Close-up views of the hydrophobic interface between α2/3$^i$ and α3$^{i+1}$ and the electrostatic interface between α1$^i$ and α2/3$^{i+1}$. Protomer (i) shown in yellow and protomer (i+1) in red. (**E**) Conceptual model for the Mup1-pHluorin MVB sorting assay. Vacuole (v). (**F**) Quantitative MVB sorting data for *snf7Δ* yeast exogenously expressing empty vector, *SNF7*, *snf7$^{L121D}$*, *snf7$^{I117E}$*, *snf7$^{M114E}$*, *snf7$^{M107E}$*, *snf7$^{T103E}$*, *snf7$^{L99K}$*, *snf7$^{M104E}$*, *snf7$^{L101E}$*, *snf7$^{A97K}$*, *snf7$^{I94E}$*, *snf7$^{Q90K}$*, *snf7$^{M87E}$*, and *snf7$^{T83E}$*. Error bars represent standard deviations. (**G**) Quantitative MVB sorting data for *snf7Δ* yeast exogenously expressing empty vectors, *SNF7*, *snf7$^{R25E\ H29E\ K36E}$* and empty vector, empty vector and *snf7$^{E95K\ E102K\ E109K}$*, and *snf7$^{R25E\ H29E\ K36E}$* and *snf7$^{E95K\ E102K\ E109K}$*. Error bars represent standard deviations. (**H**) Representative TEM images of recombinant full-length Snf7$^{R52E}$, Snf7$^{R52E\ Q90K}$, Snf7$^{R52E\ M107E}$, Snf7$^{R52E\ R25E\ H29E\ K36E}$, Snf7$^{R52E\ E95K\ E102K\ E109K}$, and Snf7$^{R52E\ R25E\ H29E\ K36E}$ and Snf7$^{R52E\ E95K\ E102K\ E109K}$ (1:1). Scale bars, 200 nm.

The following figure supplements are available for figure 4:

**Figure supplement 1.** Hydrophobic Interface Mutant Analysis.

*Figure 4 continued*

**Figure supplement 2.** Western blotting analyses of *snf7Δ* yeast expressing *SNF7*, *snf7^{L121D}*, *snf7^{I117E}*, *snf7^{M114E}*, *snf7^{M107E}*, *snf7^{T103E}*, and *snf7^{L99K}*, and *SNF7*, *snf7^{M104E}*, *snf7^{L101E}*, *snf7^{A97K}*, *snf7^{I94E}*, *snf7^{Q90K}*, *snf7^{M87E}*, and *snf7^{T83E}*.

**Figure supplement 3.** Quantitative MVB sorting data for *snf7Δ* yeast exogenously expressing empty vector, *SNF7*, *snf7^{R25E}*, *snf7^{H29E}*, *snf7^{K36E}*, *snf7^{E95K}*, *snf7^{E102K}*, *and snf7^{E109K}*, and empty vector, *SNF7*, *snf7^{R25E K36E}* and *vector, vector* and *snf7^{E95K E109K}*, *snf7^{R25E K36E}* and *snf7^{E95K E109K}*.

**Figure supplement 4.** Western blotting analyses of *snf7Δ* yeast expressing *SNF7*, *snf7^{R25E H29E K36E}*, and *snf7^{E95K E102K E109K}*.

**Figure supplement 5.** Western blotting analyses of *ex vivo* P13 fractions BMOE crosslinking by Snf7^{K35C} with Snf7^{K60C}, Snf7^{A63C}, Snf7^{K69C}, Snf7^{Q75C}, Snf7^{E81C}, Snf7^{E88C}, Snf7^{E95C}, and Snf7^{E102C}.

*figure supplement 1*). As a result, mutants M104E, L101E, A97K, I94E, Q90K, M87E and T83E showed severe sorting defects, with sorting efficiencies from 12% to 76%, and mutants L121D, I117E, M114E, M107E, T103E and L99K from 7% to 34% (*Figure 4F* and *Figure 4—figure supplement 2*). Correspondingly, we previously demonstrated that the L121D mutant blocks Snf7 polymerization *in vivo* and *in vitro*, and missorts the MVB cargo carboxypeptidase S, Cps1 (*Saksena et al., 2009*). Furthermore, recombinant Snf7^{R52E Q90K}, Snf7^{R52E I94E}, Snf7^{R52E M107E}, and Snf7^{R52E M114E} proteins were able to be purified to homogeneity, but unable to generate protofilaments visible by TEM (*Figure 4H* and *Figure 4— figure supplement 1*).

We also observed electrostatic interactions between α1 of protomer (i) and α2/3 of protomer (i+1) (*Figure 4D*). This interaction is also dependent upon the extension of α2/3, and appears to be important for the positioning of α1 in the protofilament. To validate whether these inter-protomer electrostatic interactions occur *in vivo*, we generated and tested charge-inversion mutations, *snf7^{R25E H29E K36E}* and *snf7^{E95K E102K E109K}*, which resulted in severe sorting defects of 16% and 44%, respectively. Strikingly, when co-expressing both mutants *in trans*, MVB sorting was restored to 91% (*Figure 4G* and *Figure 4—figure supplements 3* and *4*). Consistently, Glu95 has been previously indicated to be involved in Snf7 inter-protomer contacts (*Shen et al., 2014*). These results are further supported by *ex vivo* crosslinking experiments. In the Snf7 polymer-enriched P13 fraction, cysteine-substituted Lys35 (α1) can be specifically crosslinked to cysteine-substituted Glu95 (α2) or Glu102

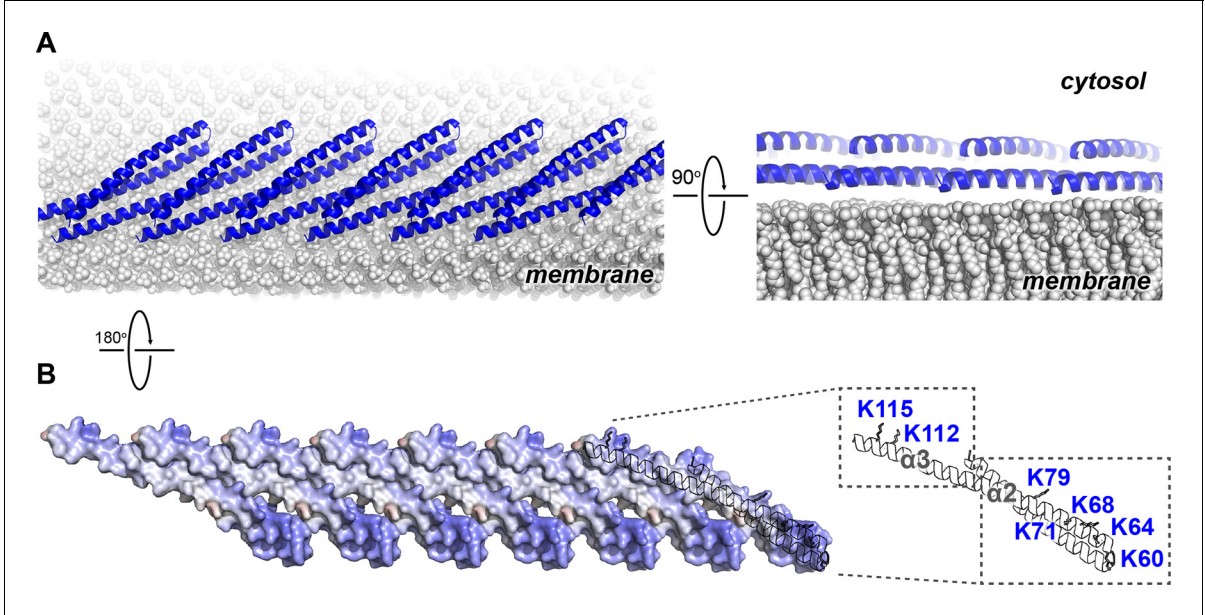

**Figure 5.** Electrostatic Protein-membrane Interactions in a Snf7 Filament (A) A Snf7 protofilament in ribbons placed on a lipid membrane in spheres (grey) (*Heller et al., 1993*). (B) Electrostatic surface potential showing the membrane interacting surface of a Snf7 protofilament with positively charged regions in blue (+10kcal/e⁻) and negatively charged regions in red (-10kcal/e⁻).

(α3) *in trans* (*Figure 4—figure supplement 5*). Furthermore, co-incubating recombinant Snf7[R52E] [R25E H29E K36E] and Snf7[R52E E95K E102E E109K] proteins resulted in protofilament formation, but no protofilaments were detected when each mutant was tested individually (*Figure 4H*).

Altogether, these *in vivo* and *in vitro* data provide strong evidence that the observed hydrophobic and electrostatic interfaces are required for Snf7 polymerization *in vivo*, and that the Snf7 protofilament observed in the crystal lattice is physiologically relevant.

## The Snf7 polymer exposes a cationic membrane-binding surface

We next mapped the previously determined Snf7 membrane-interacting region (*Buchkovich et al., 2013*) onto the Snf7 polymer structure (*Figure 5A*). Strikingly, several key conserved lysine residues, K60 K64 K68 K71 K79 (α2), and K112 K115 (α3), which were in distinct α-helices in the closed state, are arranged on an elongated and solvent-exposed surface ideal for interacting with the acidic endosomal membrane. The electrostatic membrane-binding regions of all Snf7 protomers face the same direction in the polymer, allowing for a continuous membrane-binding interface (*Figure 5B*). Thus, the crystal structure of Snf7 polymers reveals a mechanism for coupling polymerization to stable membrane association.

Notably, ESCRT-III subunits utilize multiple hydrophobic and electrostatic interfaces to interact with endosomal membranes (*Buchkovich et al., 2013*). Consistent with this, we observed that α1 of Snf7 is moderately positively charged (*Figure 1C*), and cannot rule out that at some stage of vesicle biogenesis it also comes in contact with the membrane.

## α4 bridges two Snf7 protofilaments

Comparison of our Snf7[core] crystal structures we determined revealed two distinct conformations. Although both structures exhibit an open conformation, we noted two different orientations of α4 with respect to the α1/2 hairpin. In open conformation A, α4 extends in the protofilament plane, whereas in open conformation B, α4 is positioned perpendicular to the protofilament plane. Superimposing the two conformations revealed that α4 can rotate by at least ~90° along the axis of the α2/3 helix (*Figures 6A–B*). Despite the large differences in α4 positioning, α4 makes a similar interaction with the α1/2 hairpin of a Snf7 protomer in a neighboring protofilament in both structures (*Figure 6C* and *Figure 6—figure supplement 4*). This supports a model in which the assembly of the extended α2/3 helix upon Snf7 activation results in two key events: (1) α4 can no longer bind *in cis* to its own protomer; (2) α5 is displaced from the α1/2 hairpin. Together, this enables α4 to contact the α1/2 hairpin of another protomer *in trans* on a neighboring protofilament.

Consistent with the DEER data that two Snf7 protofilaments do not make extensive contacts with each other and do not assemble into homogeneous arrangements, this interfilamental interface only buries 474 Å$^2$ of solvent-accessible surface area per protomer. To test whether these observed interfilamental interactions were functionally important, we mutated residues at their α1/2[i]-α4[j] interface (*Figures 6D–E*). Notably, Met130, which interacts with Gln90 in the closed state, is involved in this interface in the open state. Snf7 mutants of A51E, L55E, L67E, V126E, M130E and I133E led to drastic loss-of-function, with sorting efficiencies from 9% to 55% *in vivo*, and were unable to assemble into protofilaments *in vitro* (*Figure 6F* and *Figure 6—figure supplements 1* and *2*).

To gain insights into the importance of the local rearrangement of the Snf7 α3/4 loop *in vivo*, we mutated the conserved α3/4 loop residue Leu121 to Pro to constrain the rotational angle between α3 and α4. The L121P mutant exhibited a MVB sorting efficiency of 32%, compared to that of the α1/2 loop residue Asn59 mutant N59P of 75% (*Figure 6— figure supplement 3*), suggesting that the α3/4 loop functions as an important flexible 'hinge' that may facilitate different architectural stages of Snf7 polymers (*Figure 6C* and *Figure 6—figure supplement 4*).

Interestingly, studies have previously shown that the tip of the α1/2 hairpin is important for intra- and inter-molecular contacts of ESCRT-III subunits. For example, X-ray crystal structures of CHMP3 and IST1 are autoinhibited through an intramolecular contact between the α1/2 hairpin and α5 (*Figure 6—figure supplement 5*) (*Bajorek et al., 2009*; *Muziol et al., 2006*); and the Ist1-Did2 co-crystal structure revealed that the MIM1 of CHMP1B forms an intermolecular contact with the α1/2 hairpin of Ist1 (*Xiao et al., 2009*).

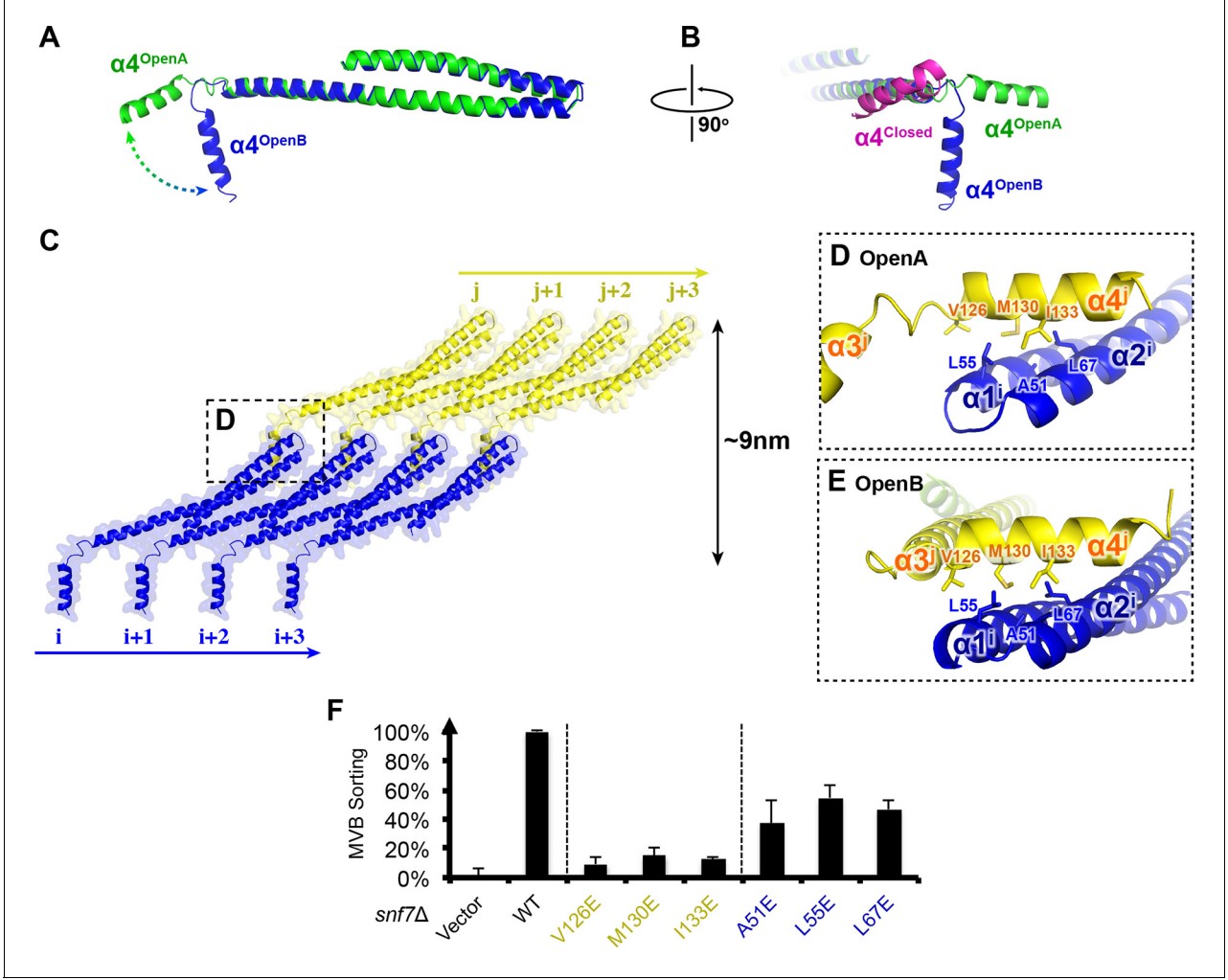

**Figure 6.** Snf7 α4 in Inter-Filament Interactions (A–B) Snf7$^{core}$ conformations A (green) and B (blue) superimposed. (B) 90° rotation and superimposing with a closed CHMP3 (purple) using its α3 as a reference. (C) Overlay of ribbon and space-filling models of the Snf7$^{core}$ crystal packing of the open conformation A. The dash-line box represents the interfilament contacts. Arrows represent inter-protofilament orientations. (D–E) Close-up views of the hydrophobic interface between α1/2$^i$ (blue) and α4$^j$ (yellow) of open conformations (D) A and (E) B. (F) Quantitative MVB sorting data for *snf7Δ* yeast exogenously expressing empty vector, *SNF7*, *snf7$^{V126E}$*, *snf7$^{M130E}$*, *snf7$^{I133E}$*, *snf7$^{A51E}$*, *snf7$^{L55E}$*, and *snf7$^{L67E}$*. Error bars represent standard deviations. See also **Table 1**.

The following figure supplements are available for figure 6:

**Figure supplement 1.** Representative TEM images of recombinant full-length Snf7$^{R52E\ V126E}$ and Snf7$^{R52E\ I133E}$.

**Figure supplement 2.** Western blotting analyses of *snf7Δ* yeast expressing *SNF7, snf7$^{A51E}$, snf7$^{L55E}$, snf7$^{L67E}$, snf7$^{V126E}$, snf7$^{M130E}$*, and *snf7$^{I133E}$*.

**Figure supplement 3.** Quantitative MVB sorting data for *snf7Δ* yeast exogenously expressing *SNF7, snf7$^{E102P}$, snf7$^{N59P}$*, and *snf7$^{L121P}$*.

**Figure supplement 4.** An overlay of ribbon and space-filling models of the Snf7$^{core}$ crystal packing of the open conformation B.

**Figure supplement 5.** Superimposing of Snf7$^{core}$ subunit (i) (blue), (j) (yellow) and CHMP3$^{α1-α5}$ (purple) of open conformations A (upper) and B (lower).

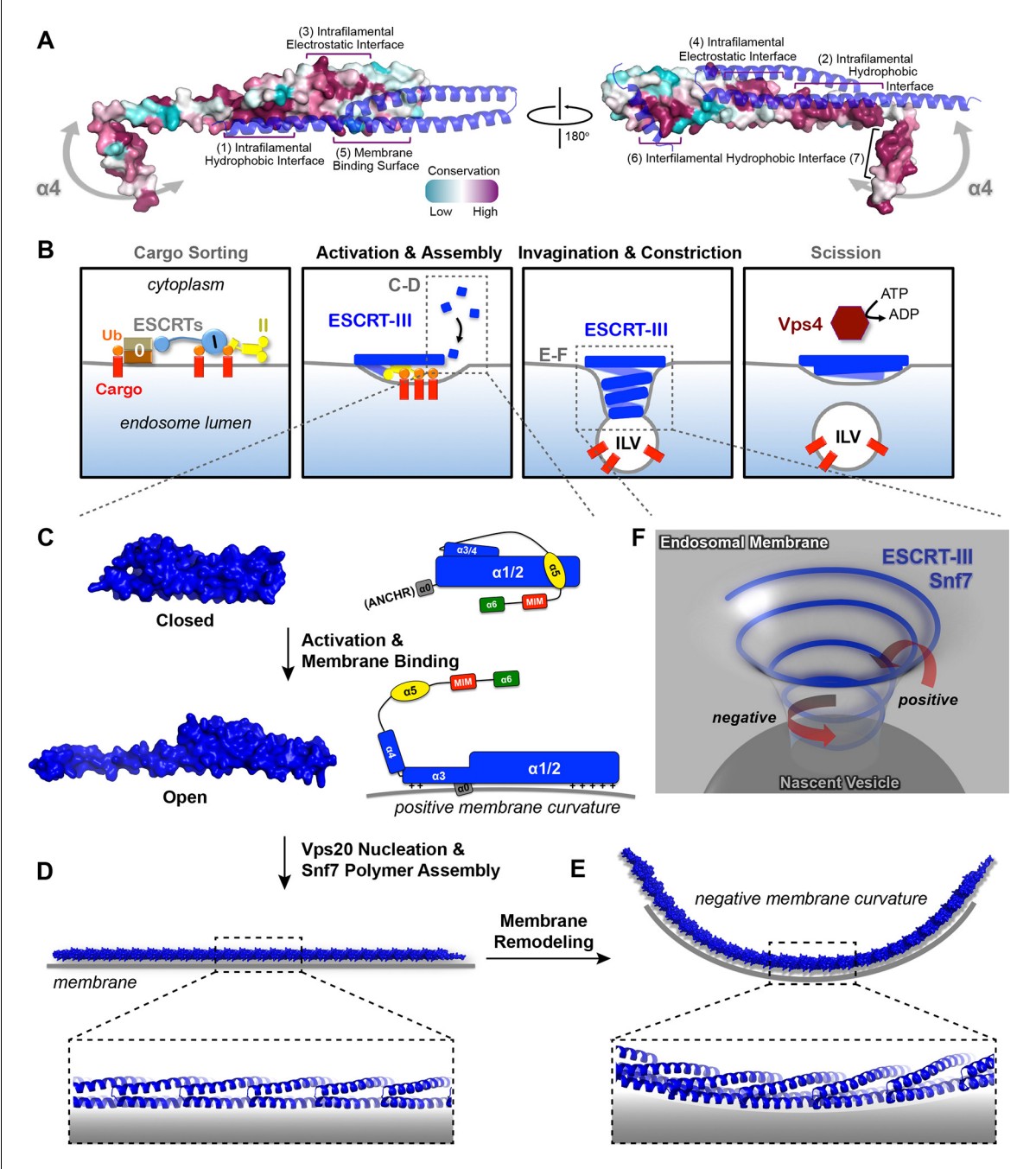

**Figure 7.** Models of Snf7 activation, polymer assembly and membrane remodeling (**A**) Space-filling CONSURF models with high conservation (purple) and low conservation (cyan). Interacting protomers shown in ribbon (blue). Seven conserved regions with assigned functions labeled. Gray arrows indicate the flexibility of α4. (**B**) Speculative cartoons illustrating four stages in ESCRT-mediated vesicle budding. (**C**) Space-filling models and schematic cartoons of Snf7core in closed and open states with membrane (grey). (**D**) Space-filling and close-up ribbon models of a 25-mer Snf7 single filament with membrane. (**E**) Space-filling and close-up ribbon models of a 23-mer Snf7 normal mode analysis filament with membrane (grey). (**F**) Schematic of a Snf7 homo-polymer in the neck of a nascent ILV with positive and negative membrane curvatures.

The following figure supplements are available for figure 7:

**Figure supplement 1.** Alignment of Snf7core protein sequences from *Saccharomyces cerevisiae (Sc), Homo sapiens (Hs), Mus musculus (Mm), Xenopus laevis (Xl), Drosophila melanogaster (Dm), Caenorhabditis elegans (Ce), Schizosaccharomyces pombe (Sp)* and *Lokiarchea (**Spang et al., 2015**)*.

**Figure supplement 2.** A ribbon model of a supercomplex of Vps25-Vps20-Snf7.

*Figure 7 continued on next page*

*Figure 7 continued*

**Figure supplement 3.** Architectures of Snf7 protofilaments

## Conserved Snf7 protein-protein and protein-membrane interfaces

To gain insights into any functionally important surfaces on the Snf7 structure, we performed CON-SURF analysis (*Celniker et al., 2013*). As a result, we identified seven highly conserved regions in the Snf7$^{core}$ domain (*Figure 7A* and *Figure 7— figure supplement 1*). Strikingly, all of them map to regions of Snf7 assigned specific functions in either polymer assembly or membrane interaction: regions (1) and (2) are located on opposite sides of the extended α2/3 helix and stabilize intrafilamental protein-protein interactions; region (3) is located towards the N-terminus of α1 and region (4) towards the middle of the α2/3 helix, forming the intrafilamental electrostatic interacting surfaces; region (5) corresponds to the beginning of α2, which we previously identified as a cationic membrane-binding surface; regions (6) and (7) are the tip of the α1/2 hairpin and the hydrophobic side of α4, which together stabilize interfilamental interactions. Thus, the Snf7 protein-protein interactions revealed from our X-ray crystal structures and the protein-membrane interactions previously identified (*Buchkovich et al., 2013*) are evolutionarily conserved.

## A linear Snf7 filament transformed into a superhelix

A linear Snf7 filament has a simple two-dimensional geometry, and thus is incapable of mediating the drastic three-dimensional membrane remodeling required for membrane deformation and vesicle formation at the endosome (*Figure 7B*). Since we know that Snf7 can form spirals on a membrane surface, we asked how a linear Snf7 filament (*Figure 7D* and *Figure 7—figure supplement 3*) could be transformed into a circular array. Due to the heterogeneity of a Snf7 double filament, we utilized a single linear Snf7 filament to determine a plausible curved Snf7 filament using normal mode analysis (NMA), a simple non-detailed simulation technique used to probe large-scale macromolecular motions by assessing flexibility intrinsic to the structure of a protein (*Suhre and Sanejouand, 2004*). Remarkably, without any dramatic structural rearrangement within each protomer or alterations of the protein-protein interface, a linear two-dimensional Snf7 filament can bend into a ~70 nm diameter three-dimensional superhelix with turn length of ~62 nm, reminiscent of the structure of the Snf7/Vps24/Vps2 co-assembly previously observed (*Henne et al., 2012*) (*Figure 7—figure supplement 3*). Importantly, this 3D helical array aligns the cationic membrane-binding surfaces on the outside of the superhelix, ideal for a Snf7 polymer to stabilize a negatively curved membrane surface (*Figure 7E*).

## Discussion

The ESCRT-III machinery plays a critical role in numerous fundamental cellular processes, including MVB biogenesis, viral budding and cytokinesis, indicating an ancient and conserved membrane remodeling mechanism. The importance of understanding this mechanism is bolstered by the fact that this is so distinct from all other well characterized membrane budding processes (*e.g.* clathrin and COP-I/-II), which invariantly bud into the cytoplasm. Although the membrane-bound ESCRT-III polymers have been reconstituted *in vitro*, the mechanisms governing the polymer assembly and how ESCRT-III coordinates membrane remodeling, remain poorly understood. Here, we focused on Snf7, the predominant ESCRT-III component, in order to fundamentally understand how it achieves membrane remodeling. Using a multi-disciplinary approach that combined X-ray crystallography, PDS, genetics, biochemistry and TEM, we reveal key structural features of Snf7 that allow its dynamic conversion from a soluble monomer to a membrane-bound polymer.

Specifically, we provide, for the first time, atomic-resolution structures of a conformationally 'open' and assembled ESCRT-III subunit, and report the first application of PDS to characterize the internal organization of protein polymers assembled on a near-native lipid environment. These Snf7 subunits assemble into remarkable linear arrays upon conformational activation, thus providing a structural explanation for the Snf7 spiraling protofilaments previously observed by TEM and mechanistic insights into the spiral-mediated membrane deformation and vesicle formation.

## Snf7 core domain rearrangement required for protein-membrane and protein-protein interactions

A classic model of ESCRT-III activation involves the disruption of intramolecular interactions between α5 and the core domain (*Henne et al., 2012*; *Lata et al., 2008a*). In the present study, we provide surprising structural evidence that this activation requires further rearrangement within the core domain itself. Consistent with the available CHMP4B$^{\alpha1/2}$ structure (*Martinelli et al., 2012*), α1/2 folds into a rigid ~70 Å α-helical hairpin, which forms intramolecular contacts with at least three short α-helices, α3, α4, and α5 in the closed state. Notably, upon activation, all of these interactions are reorganized to extend the hairpin to a ~90 Å structure available for intermolecular contacts. Comparison of the closed and open states reveals that α4 is displaced by ~60 Å as the molecule opens. Intriguingly, a recent small-angle X-ray scattering (SAXS) study showed that Vps20 exists as a 94 Å extended 'open' conformation but it is incapable of homo-polymerization (*Schuh et al., 2015*).

Based on these structural insights, we propose a detailed 'lifecycle' of Snf7 activation and polymerization: 1) In the cytoplasm, Snf7 exists in a dynamic equilibrium of mixed intermediates between the open and closed states; 2) on endosomes, Vps20 α1 directly associates with the ESCRT-II subunit Vps25 (*Im et al., 2009*), allowing Vps20 to function as a nucleator to engage an open Snf7 from the cytoplasm (*Figure 7—figure supplement 2*); 3) the open conformation of Snf7 with an extended α2/3 helix presents a cationic membrane-interacting surface to orient itself on endosomes; 4) the N-terminal membrane ANCHR motif further stabilizes Snf7 on the endosomal surface (*Figure 7C*); 5) the endosomal recruitment shifts the conformational equilibrium and thus triggers a 'domino effect' of Snf7 opening and promotes Snf7 polymerization into a ~30 Å periodic array of ordered inter-protomer contacts. In agreement with this, an averaged 32.5 Å inter-subunit distance was observed in *C. elegans* Vps32 spirals (*Shen et al., 2014*).

Because X-ray crystal structures of both Vps24 (*Muziol et al., 2006*) and Ist1 (*Bajorek et al., 2009*) were determined in their autoinhibitory conformations with an unresolved 'linker' between the core and α5, the four-helix core domain has been treated as a rigid body that remains unaltered between the open and closed states. However, a previous SAXS study suggested that Vps24 can adopt both a 75 Å globular and a 105 Å extended conformation (*Lata et al., 2008a*), implying that the core domain extension may be a common theme of ESCRT-III activation. Due to this unexpected conformational change, previous ESCRT-III polymer studies using the 'closed' conformation as a building unit may need careful reevaluation.

## Comparison of ESCRT-III filaments with other membrane-remodeling polymers

The ESCRT-III Snf7 filament-mediated membrane remodeling is conceptually reminiscent of other membrane remodeling machinery, including bacterial FtsZ (*Osawa et al., 2008*). Interestingly, both Snf7 and FtsZ/FtsA can drive cytokinetic abscission, and they share at least three distinct structural characteristics: electrostatic protein-membrane interactions, membrane insertion of an amphipathic helix, and oligomeric protein scaffolding.

Despite these similarities, the major difference between FtsZ and Snf7 is that FtsZ requires nucleotide hydrolysis to drive its conformational dynamics. The propagation of conformational changes in the FtsZ polymer is thus coupled with the architectural changes that promote membrane fission. In contrast, ESCRT-III does not bind nor hydrolyze nucleotides to regulate its conformation, but it recruits the AAA-ATPase Vps4 for its disassembly. Although Snf7 can be activated by specific point mutations *in vitro*, the conformational switching *in vivo* appears to be tightly regulated by other ESCRT components to prevent pre-mature polymer assembly.

During MVB biogenesis, ESCRT-II binds two copies of Vps20, which then nucleates the homo-oligomerization of Snf7. However, in enveloped viral budding and cytokinetic abscission, Bro1/Alix directly bridges ESCRT-I to ESCRT-III, by binding to the C-terminal α6 of Snf7 (*McCullough et al., 2008*). We speculate that this interaction may directly dissociate the C-terminal autoinhibitory region to trigger Snf7 polymer assembly. Furthermore, CHMP7 was recently shown to trigger Snf7 assembly during nuclear envelope reformation (*Vietri et al., 2015*), highlighting the distinct spatial and temporal regulation of Snf7 activation between different ESCRT-dependent processes.

## A curved ESCRT-III filament mediates membrane remodeling

ILVs that bud into the endosomal lumen contain no outer vesicle coat, yet show consistent diameters, suggesting ESCRTs regulate vesicle size. Somewhat paradoxically, ESCRT-III cannot shape the vesicle exterior because it is segregated in the cytoplasm by the limiting membrane of the endosome. Instead, the ESCRT-III filament appears to predominantly drive membrane deformation by sculpting the neck interior of a growing vesicle. This membrane sculpting requires an intricate balance of competing curvatures. Snf7 has been shown to localize to both the curved necks of invaginations and along highly curved edges of membranes (*Buchkovich et al., 2013*; *Fyfe et al., 2011*; *Wollert and Hurley, 2010*) (*Figure 7F*). Our collective studies on Snf7 address the balance of membrane curvatures associated with ILV formation. The ANCHR motif of Snf7 acts to sense and stabilize the positively curved rim of the invagination (*Figure 7C*). Coinciding with this positive curvature stabilization, the helical Snf7 polymer acts as a circular scaffold that triggers and stabilizes the negatively curved circumference of the neck of the invagination (*Figure 7E*). We propose that as a two-dimensional ESCRT-III spiral elongates into a three-dimensional superhelix, the tight membrane binding of the 'corkscrew' concentrates transmembrane cargoes ahead of the leading edge of the forming and narrowing filament, packaging them into the nascent ILV (*Figure 7B*).

Despite the reconstitution and high-resolution analysis of Snf7 polymers, key questions remain. The most pressing are the mechanisms governing inter-ESCRT-III subunit interactions, particularly, Vps24 and Vps2, required for the ESCRT-III architectural changes, and a precise mechanochemical role of the AAA-ATPase Vps4 complex during the final membrane constriction and scission coupled with ESCRT-III disassembly. Additional structural studies together with new assays are necessary for further addressing these challenging but exciting questions.

## Materials and methods

### Protein crystallization

The DNA sequence encoding *Saccharomyces cerevisiae* Snf7$^{core}$ (residues 12–150) was subcloned into a pET28a vector with an N-terminal His$_6$-Sumo tag. Recombinant proteins were overexpressed in *Escherichia coli* Rosetta cells and purified by TALON metal affinity resin. The His$_6$-Sumo tag was removed by Ulp1 protease at 4°C overnight. The mixture was further purified by Superdex-200 gel filtration. The peak corresponding to Snf7$^{core}$ was pooled and concentrated in a buffer of 300 mM NaCl, 20 mM HEPES pH7.4.

Snf7$^{core}$ (conformation A) was crystallized in a hanging-drop vapor diffusion system at 4°C by mixing protein (5.7 mg/mL) with reservoir solution containing 100 mM NaCl, 100 mM MES:NaOH pH5.5, 3% PEG20,000 in 1:1 ratio (*v/v*). Crystals were transferred into the same solution supplemented with 30% glycerol before cooling to liquid nitrogen temperature under atmosphere pressure. Snf7$^{core}$ (conformation B) crystals were grown in 110 mM NaCl, 70 mM MES:NaOH pH5.5, 6% PEG20,000 and subject to high-pressure cryo-cooling (*Kim et al., 2005*). The crystals were mounted in oil on a pin with a piece of steel piano wire attached to the base, pressurized to 200MPa and cooled to liquid nitrogen temperature. The pressure was then released while keeping the temperature unaltered.

### X-ray crystallography

X-ray diffraction data was collected on Snf7 crystal 'A' to 2.4 Å at MacCHESS beam line F1 of Cornell High Energy Synchrotron Source. The crystal belonged to space group $P2_1$ with unit cell dimensions $a$=29.5 Å $b$=52.2 Å $c$=54.5 Å $\alpha$=90° $\beta$=97.5° $\gamma$=90°. X-ray diffraction data was collected on crystal 'B' to 1.6 Å. It belonged to space group $P2_1$ with unit cell dimensions $a$=29.9 Å $b$=46.2 Å $c$=44.6 Å $\alpha$=90° $\beta$=98.5° $\gamma$=90°. Diffraction data were processed using *HKL-2000* (*Otwinowski and Minor, 1997*). There is one Snf7 molecule in the asymmetric unit of both crystals. The structures were solved using *Phaser* in *Phenix* (*Adams et al., 2010*) by molecular replacement with CHMP4B$^{\alpha1-\alpha2}$ (PDB: 4ABM) as a search model. Refinement and density modification were performed in *Phenix*. Model building was performed using *Coot* (*Emsley and Cowtan, 2004*). Throughout this study, structural images were generated with *PyMOL* using the 1.6 Å structure unless otherwise noted.

## Protein purification

All Snf7 protein purification for PDS and TEM analyses were performed as previously described (*Henne et al., 2012*). Briefly, Snf7 constructs were subcloned into a pET23d bacterial expression vector (Novagen) with an N-terminal His$_6$-tag. Recombinant proteins were overexpressed by *Escherichia coli* BL21 or C41 cells, purified by TALON metal affinity resin and eluted in 150 mM NaCl, 20 mM HEPES pH7.4 and 400 mM Imidazole. The elution fractions were pooled and further purified by Superdex-200 gel filtration in a buffer of 150 mM NaCl, 20 mM HEPES pH7.4.

## Site-directed spin-labeling

Recombinant Snf7 cysteine-substituted proteins were purified and enriched on TALON resin, and spin-labeled with 1 µg/mL *S*-(1-*oxyl*-2,2,5,5-tetramethyl-2,5-dihydro-1*H*-pyrrol-3-yl)methyl methanesulfonothioate, MTSL (Santa Cruz Biotech) dissolved in acetonitrile at 4°C overnight. The spin-labeled proteins were eluted in 150 mM NaCl, 20 mM HEPES pH7.4, 400 mM Imidazole and further purified by Superdex-200 gel filtration in a buffer of 150 mM NaCl, 20 mM HEPES pH7.4 to remove unreacted spin labels.

## Sample preparations for DEER sectroscopy

For soluble protein samples, spin-labeled proteins were buffer exchanged in a 10 kDa molecular weight cutoff protein concentrator (Millipore) to ~80% deuterium buffer of 150 mM NaCl, 20 mM HEPES pD7.4 supplemented with 30% (*v/v*) glycerol-d$_8$. For liposome-reconstituted protein samples, 1 mg/mL of 800 nm diameter 60% 1,2-dioleoyl-*sn*-glycero-3-phosphocholine (DOPC), 30% 1,2-dioleoyl-*sn*-glycero-3-phospho-*L*-serine (DOPS), 10% phosphatidylinositol 3-phosphate (PI(3)P) liposomes were generated as previously described (*Buchkovich et al., 2013*). 25 µL of 10-30 µM proteins and 25 µL 1 mg/mL liposomes were coincubated at room temperature for 15 min and ultracentrifuged in a TLA-100 rotor (Beckman Coulter) for 10 min at 70,000 rpm at 20°C. A total of 6 liposome pellets were combined and resuspended in 20 µL deuterium buffer of 150 mM NaCl, 20 mM HEPES pD7.4 supplemented with 15% (*v/v*) glycerol-d$_8$, resulting in a sample of ~10-30 µM protein: ~3 mg/mL lipid for DEER measurements.

## DEER data collection and analysis

20 µL spin-labeled samples were loaded into 1.8 mm inner diameter Pyrex sample tubes (Wilmad-LabGlass), shock frozen in liquid nitrogen prior to DEER measurements. DEER measurements were performed at 60 K using a home-built Ku band 17.3 GHz pulse electron spin resonance spectrometer (*Borbat et al., 1997*; *Borbat et al., 2013*). A four-pulse DEER sequence (*Jeschke and Polyhach, 2007*) was used routinely with the detection π/2- and π-pulses having widths of 16 and 32 ns and pump π-pulse of 16 ns. The detection pulse sequence was applied at the low-field spectral position, while pumping was performed at a lower by 70 MHz frequency positioned at the central maximum. A 32-step phase cycle (*Gamliel and Freed, 1990*) was applied to suppress unwanted contributions to the signal. Nuclear modulation effects caused by surrounding protons were suppressed by averaging the data from 4 measurements with slightly different separations of the first two pulses, *i.e.* advanced by 9.5 ns for subsequent measurement. Depending on spin-labeled protein concentration, distance, and phase relaxation time, DEER data were usually acquired in less than 12 hr.

Time-domain DEER data, *V(t)*, were reconstructed into distance distributions using standard approaches (*Borbat and Freed, 2007*; *Borbat and Freed, 2014*; *Jeschke, 2012*; *Jeschke and Polyhach, 2007*). First, the signal decay due to intermolecular spin interactions was removed from *V(t)* by approximating the latter points (about a half of the record) of ln*V(t)* with a low-order polynomial, usually nearly a straight line, and subtracting it out from ln*V(t)* so that the antilog yields *u(t)*. Once normalized as $V(t) = \frac{u(t)}{u(0)}$, it serves as a typical form of DEER data presentation, while *u(t)*-1 gives background free data, which was subsequently converted to a distance distribution between spin pairs with L-curve Tikhonov regularization (*Chiang et al., 2005a*) followed, when needed, by maximum entropy method refinement (*Chiang et al., 2005b*). The modulation depth, defined as 1-*V*(∞), where *V*(∞) is the asymptotic value of *V(t)*, was used to report on the presence and extent of multi-spin effects (*Bode et al., 2007*).

For mapping Snf7 conformation, we employed double spin-labeled Snf7 and magnetic dilution (*Borbat and Freed, 2007*; *Dzikovski et al., 2011*; *Meyer et al., 2014*; *Pornsuwan et al., 2013*).

*Figure 3—figure supplement 1* demonstrates a benchmark magnetic dilution study of double-labeled Snf7$^{R52E\ K60C\ A66C}$ with unlabeled Snf7$^{R52E}$. This spin pair at the tip of the α1/2 hairpin was selected as a reference for inspecting bound protein conformational variability and the conditions for isolation of intramolecular distances. The distance of this construct in solution is ~20 Å, in agreement with spin-label modeling into a homology structure (*Figure 2A*) using *MMM (m*olecular *m*ulti-scale *m*odeling) software package (*Polyhach et al., 2011*). Generic MTSL rotamer library for 298 K was used to determine conformations of attached spin labeled cysteine side chains and produce distance distributions between pairs of labeled sites. Distance distributions FWHMs were in the range of 0.4-1.2 nm. Respective background free time-domain data were generated with the help of the same package.

Consistently, Snf7 was found to be structurally more heterogeneous in solution, producing broad distributions based on DEER data for a set of double-labeled Snf7 full-length constructs (*Figure 2—figure supplement 3*). Intriguingly, this study revealed a distinct Snf7 conformation in the protofilaments, which manifests itself as a very narrow distance distribution already at mild magnetic dilution (1:2), thus pointing to a low extent of intermolecular contacts. In the absence of unlabeled proteins (1:0 magnetic dilution), Snf7$^{R52E\ K60C\ A66C}$ in liposome samples produced broad distributions, which showed a range of distances to neighbors with ~30 Å being dominant (*Figure 3—figure supplement 1*). In addition, the large modulation depth indicated coupling to at least two neighbors. This indicated that for isolating longer distances considerably higher dilution ratios would be desirable.

*Figures 2I,K* and *Figure 3—figure supplement 2* illustrate subsequent application of this method to the membrane-bound Snf7$^{R52E\ E88C\ H118C}$. Note that in *Figure 2I*, the reconstructed distance distribution of soluble Snf7$^{R52E\ E88C\ H118C}$ is normalized at a 4x scale than the MMM simulation data to illustrate the structural heterogeneity. In *Figure 2K*, the reconstructed distance distributions have a large component of ~30 Å originating from distances to immediate neighbors similar to the benchmark case, the magnetically diluted samples have a single peak at ~45 Å that is dominant with only a small fraction at 30 Å that could still be noticed at 1:8 dilution. A dilution factor in excess of 15 would be necessary to fully reveal the expected signal shape, however the 1:8 dilution sample already has ~5 µM protein concentration, making larger ratios problematic to study.

Snf7 polymeric packing was assayed by inspecting intermolecular dipolar couplings for various single-labeled constructs assembled in protofilaments on liposome membranes. Whereas the most pronounced distance is expected to be determined by the proximal neighbors, the widths of distance distributions (*Figures 3B–E*) obtained in these scans are likely to have contributions from the couplings to more distant neighbors and in addition by the complex nature of the Snf7 polymer in a liposome-reconstituted system where Snf7 spiraling double- protofilaments are observed, and the orientation relative to each other is heterogeneous (*Cashikar et al., 2014*; *Henne et al., 2012*). Notably, while searching for 'tip-to-tip' contacts possible in double protofilaments, we did not identify any spin labeled position with a distinct short proximity that is expected to occur at the contacting edge of the single filament, thus ruling out this scenario. We also did not discern any significant distance variation as the spin labeled position is moved from one end of the α1-3 hairpin to the other, thus ruling out any alternating protomer packing in the protofilaments. So far, only parallel protomer packing in a single-layer filament is consistent with the data (see also Result).

## Subcellular fractionation analysis

Subcellular fractionation experiment was performed as previously described (*Buchkovich et al., 2013*). Briefly, 30 OD$_{600nm}$V of mid-log yeast cultures were spheroplasted in Zymolyase and lyzed in 1 mL of 50 mM Tris pH7.4, 1 mM EDTA, 200 mM sorbitol with protease inhibitors (Roche). Lysates were cleared at 500 x*g* for 5 min at 4°C, and then fractionated by centrifugation at 13,000 x*g* for 10 min at 4°C. The supernatant (S13) fraction was collected. The pellet (P13) fraction was resuspended in 1 mL lysis buffer. Both fractions were then precipitated by 10% trichloroacetic acid for at least 30 min and washed by acetone twice.

## Dicysteine oxidative crosslinking

The oxidizing chemical copper(II) 1,10-phenanthroline was prepared freshly. 9 mg copper(II) sulfate was dissolved in 250 µL ionic buffer of 150 mM postasium acetate, 5 mM magnesium acetate, 250 mM sorbitol, 20 mM HEPES pH7.0. 20 mg 1,10-phenanthroline was dissolved in 500 µL ethanol.

Both solutions were mixed creating a brilliant aqua-colored solution with white precipitate. 7 μL copper(II) 1,10-phenanthroline solution was added into 450 μL of S13 or P13 fractions, and incubated at 4°C for 15 min. Samples were then precipitated by 10% trichloroacetic acid, washed twice by acetone and subjected for western blotting analysis.

### Ex vivo dicysteine crosslinking by crosslinker

30 $OD_{600nm}$V of mid-log yeast cultures were spheroplasted, lyzed and fractionated. The 1 mL P13 fractions were equally divided into two subfractions. Subfraction 1 was treated with 20 μL DMSO and subfraction 2 with 20 μL 20 mM bismaleimidoethane (BMOE) (Life Technologies) in DMSO for 2hours at 4°C. Excessive BMOE was quenched by adding 0.2 μL 1 M dithiothreitol. Samples were then precipitated by 10% trichloroacetic acid, washed twice by acetone and subjected for western blotting analysis.

### Flow cytometry, microscopy, western blotting, yeast strain and plasmids

The quantitative Mup1-pHluorin ESCRT cargo-sorting flow cytometry assay, negative stain TEM, and western blotting were performed as previously described (*Buchkovich et al., 2013*; *Henne et al., 2012*). See *Supplementary file 1* for a list of plasmids and yeast strains used.

### CONSURF analysis

*Saccharomyces cerevisiae* Snf7 protein sequence was input as a query sequence for a protein BLAST analysis using the algorithm of blastp (protein-protein BLAST). The top 100 sequences from the result were subjected for ClustalW sequence alignment. The multiple sequence alignment and the Snf7 conformation B structure were then used as input for conservational analysis using the *CONSURF* server (*Ashkenazy et al., 2010*; *Berezin et al., 2004*; *Celniker et al., 2013*). The overall conservation scores calculated using the Bayesian method were color-coordinately mapped onto the Snf7 structure shown in *Figure 7A*.

### Normal mode analysis

Calculation of the normal modes of the Snf7 polymer was preformed on the *elNémo* server (*Suhre and Sanejouand, 2004*), by using a 25-mer of Snf7 of conformation B as an input structure. To model a circular structure with a diameter of ~65–70 nm, perturbation parameters of DQMIN of -10000, DQMAX of 10000, and DQSTEP of 2000 were applied. This yielded 3 nontrivial normal modes numbered 7, 8 and 9. The lowest frequency nontrivial normal mode, mode 7, was used. Using *Coot*, the middle 12 protomers of the No.7 normal mode were selected and then superimposed in a head-to-tail fashion to manually generate a 23-mer and 94-mer shown in *Figures 7E* and *Figure 7—figure supplement 3*.

### Accession number

Coordinates and structure factors for Snf7[core] have been deposited in the RCSB Protein Data Bank (http://www.rcsb.org) under accession PDB ID 5FD7 (open conformation A) and 5FD9 (open conformation B).

## Acknowledgements

We gratefully thank Kim YJ, Griffin ST, Yeh Y, Timashev LA and Georgieva ER for all technical expertise. We thank Cornell High Energy Synchrotron Source MacCHESS beam line F1 staff for X-ray crystallographic data collection, and Huang Q for the high-pressure cryo-cooling. We thank Chappie JS and Li M for critical reading of the manuscript, and Crane BR, Brown WJ and Zhang L for helpful discussion.

## Additional information

### Funding

| Funder | Grant reference number | Author |
|---|---|---|
| Cornell University | Harry and Samuel Mann Outstanding Graduate Student Award | Shaogeng Tang |
| National Institutes of Health | NIGMS Predoctoral Training Grant in Cellular and Molecular Biology, T32GM007273 | Shaogeng Tang |
| Cornell University | Sam and Nancy Fleming Research Fellowship | W Mike Henne |
| American Cancer Society | Postdoctoral Fellowship, PF-12-062-01-DMC | Nicholas J Buchkovich |
| National Institutes of Health | NIGMS Grant, P41GM103521 | Jack H Freed |
| National Institutes of Health | NIBIB Grant, R01EB003150 | Jack H Freed |
| National Institutes of Health | NIGMS Grant, R01GM094347 | Yuxin Mao |
| National Institutes of Health | NIGMS Grant, R01GM098621 | J Christopher Fromme |
| Cornell University | Research Grant, CU3704 | Scott D Emr |

The funders had no role in study design, data collection and interpretation, or the decision to submit the work for publication.

### Author contributions

ST, PPB, Conception and design, Acquisition of data, Analysis and interpretation of data, Drafting or revising the article, Contributed unpublished essential data or reagents; WMH, YM, SDE, Conception and design, Analysis and interpretation of data, Drafting or revising the article; NJB, Conception and design, Acquisition of data, Analysis and interpretation of data, Contributed unpublished essential data or reagents; JHF, Conception and design, Drafting or revising the article; JCF, Conception and design, Acquisition of data, Analysis and interpretation of data, Drafting or revising the article

## Additional files

### Supplementary files

• Supplementary file 1. Plasmids and yeast strains used in this study. A list of plasmids for *Saccharomyces cerevisiae* expression, *Escherichia coli* expression for protein purification, and *Saccharomyces cerevisiae* strains.

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
