## [Decision Letter]

Thank you for submitting your work entitled "Structural Basis for Activation, Assembly and Membrane Binding of ESCRT-III Snf7 Filaments" for consideration by *eLife*. Your article has been favorably evaluated by John Kuriyan (Senior editor) and three reviewers, one of whom is a member of our Board of Reviewing Editors.

The following individual involved in review of your submission has agreed to reveal his identity: Alexey Merz (peer reviewer).

The reviewers have discussed the reviews with one another and the Reviewing Editor has drafted this decision to help you prepare a revised submission.

Summary:

The manuscript of Emr and colleagues is a remarkable and elegant investigation into the activation and polymerization of the Snf7 ESCRT-III to drive membrane deformation in MVB formation. Two crystal structures of the Snf7 core reveal that a loop that connects two helices in the inactive form becomes helical to produce a single long α-helix. The crystal packing suggests a model for the polymer structure that is elegantly confirmed on membranes using site-directed spin labeling and DEER spectroscopy, as well as mutational analysis of inter-filament interfaces. The authors have carefully tested hypotheses generated from the crystal structure to come to a plausible model for the nature of the filament assembly on membrane surfaces. Combined with previous work from the Emr group and others, the manuscript provides an unprecedented detailed insight into the assembly of ESCRTs in the process of MVB generation. The study is likely to be a landmark in the field.

Essential revisions:

1) The authors should validate that at least some of the hydrophobic position mutants used in the sorting assay can be expressed and purified. The implicit assumption is that the single point mutants do not affect the global fold but only the interface, but it is not clear that this is indeed the case. Currently the only analysis is by Westerns, which does not exclude being completely misfolded. Even if the yield of such preps does not match the WT, it would still indicate that folded protein can be obtained.

2) The structural data and analysis are well described, but for the general reader it would help to explain how normal mode analysis provides the alternative conformation that leads to the curved structure used to bend membranes. Since Vps20 nucleates oligomerization of Snf7, the authors might comment on whether Vps20 might stabilize this alternative conformation and thereby promote the spiral structure. Also, the inter-subunit contacts characterized here should be explicitly compared to the h2-h2 interactions inferred by Shen et al. (2014).

---

## [Author Response]

Essential revisions:

*1) The authors should validate that at least some of the hydrophobic position mutants used in the sorting assay can be expressed and purified. The implicit assumption is that the single point mutants do not affect the global fold but only the interface, but it is not clear that this is indeed the case. Currently the only analysis is by Westerns, which does not exclude being completely misfolded. Even if the yield of such preps does not match the WT, it would still indicate that folded protein can be obtained.*

In addition to showing that these mutants can be expressed at levels essentially identical to expression of the wild-type protein *in vivo* by Western blotting, we have successfully purified several of these ‘hydrophobic position’ mutants *in vitro* and used them for transmission electron microscopy (TEM) assays shown in Figure 4 and Figure 4—figure supplement 1. We now include size-exclusion chromatograms of these mutants, showing that they eluted as a monodispersed peak, at the same retention volume compared to the Snf7^R52E^ proteins (Figure 8). This indicates that these mutants can be folded properly and purified for biochemical assays. It should be noted that these mutants also behaved similarly to the wild-type protein during their purification. Also, Snf7^L121D^ has been purified and analyzed for Snf7 polymerization in Saksena et al. 2009. To address this concern, we have modified a sentence in the Results: “Furthermore, recombinant Snf7^R52EQ90K^, Snf7^R52EI94E^, Snf7^R52EM107E^, and Snf7^R52EM114E^ proteins were able to be purified to homogeneity, but unable to generate protofilaments visible by TEM (Figure 4 and Figure 4—figure supplement 1).”

Author response image 1.Size Exclusion Chromatograms of full-length Snf7^R52E^, Snf7^R52E I94E^ and Snf7^R52E M107E^ by Superdex-200 gel filtration.**DOI:**
http://dx.doi.org/10.7554/eLife.12548.035

*The structural data and analysis are well described, but for the general reader it would help to explain how normal mode analysis provides the alternative conformation that leads to the curved structure used to bend membranes.*

We have modified a sentence in the subsection “A Linear Snf7 Filament Transformed into A Superhelix “to further explain the normal mode analysis methodology. We included the detailed procedures we used for this analysis in the Materials and methods section.

*Since Vps20 nucleates oligomerization of Snf7, the authors might comment on whether Vps20 might stabilize this alternative conformation and thereby promote the spiral structure.*

A previous study from our lab (Henne et al.2012) suggested that Snf7/Vps24/Vps2 can co-assemble into such a 3-dimensional spiral structure for membrane bending. But Snf7 co-assembles with Vps20 and ESCRT-II into a 2-dimensional ring structure. As included in the Discussion, as Vps20 adopts a 94Å extended open conformation (Schuh et al., 2015) and anchors to the endosomes by myristoylation, we propose that an open Vps20 would stabilize a Snf7 in its open conformation on the membrane and thus trigger a chain reaction of Snf7 recruitment and polymerization. Following this, the recruitment of Vps24/2 alters the architectural geometry into a membrane-bending spiral.

*Also, the inter-subunit contacts characterized here should be explicitly compared to the h2-h2 interactions inferred by Shen et al. (2014).*

We agree with the reviewers that it would be useful to compare the α2-α2 interactions in our study with the proposed Snf7 packing model in Shen et al. 2014. Unfortunately, we are unable to access the coordinates from the Shen et al. study, and thus unable to superimpose these structures for detailed structural analysis. Despite this, we have noted that a similar ~30Å inter-subunit distance has been previously observed by Shen et al. Moreover, Glu95 (α2), which has been suggested as a key residue for inter-subunit contact by Shen et al., is further confirmed by our crystal structures. We have added one sentence to connect our study to this previous study: “Consistently, Glu95 has been previously indicated to be involved in Snf7 inter- protomer contacts (Shen et al., 2014).”